# In Vitro Antibacterial and Antibiofilm Activity of Hungarian Honeys against Respiratory Tract Bacteria

**DOI:** 10.3390/foods10071632

**Published:** 2021-07-14

**Authors:** Viktória Lilla Balázs, Lilla Nagy-Radványi, Rita Filep, Erika Kerekes, Béla Kocsis, Marianna Kocsis, Ágnes Farkas

**Affiliations:** 1Department of Pharmacognosy, Faculty of Pharmacy, University of Pécs, 7624 Pécs, Hungary; balazsviktorialilla@gmail.com (V.L.B.); radvanyililla25@gmail.com (L.N.-R.); rita.filep@gmail.com (R.F.); 2Department of Microbiology, Faculty of Science and Informatics, University of Szeged, 6720 Szeged, Hungary; kerekeserika88@gmail.com; 3Department of Medical Microbiology and Immunology, Medical School, University of Pécs, 7624 Pécs, Hungary; kocsis.bela@pte.hu; 4Institute of Biology, Faculty of Sciences, University of Pécs, 7624 Pécs, Hungary; mkocsis@gamma.ttk.pte.hu

**Keywords:** *Pseudomonas aeruginosa*, *Haemophilus* spp., *Streptococcus pneumoniae*, antibiofilm activity, antibacterial effect, honey, respiratory tract

## Abstract

Honey is a rich source of carbohydrates, while minor compounds such as amino acids and polyphenols contribute to its health-promoting effects. Honey is one of the oldest traditional remedies applied for microbial infections, due to its antibacterial, anti-inflammatory, and antioxidant properties. The aim of this study was to investigate the antibacterial and antibiofilm effects of Hungarian black locust, linden, and sunflower honeys against the most common biofilm-forming respiratory tract pathogens *Haemophilus* spp., *Pseudomonas aeruginosa*, and *Streptococcus pneumoniae*. The unifloral character of all three honey types was confirmed by melissopalynological analysis. The antibacterial activity of each honey sample against each bacterium strain was proven with agar well diffusion assay and thin layer chromatography—direct bioautography. Kinetics and mechanisms of antibacterial action were clarified with time-kill assay and membrane degradation study. The anti-biofilm activity was evidenced using crystal violet assay. In each assay, linden honey was the most effective, followed by sunflower and black locust honey. In addition, each honey sample had greater potential to suppress respiratory tract bacteria, compared to major sugar components. In conclusion, honey in general and linden honey in particular, can have a role in the treatment of respiratory tract infections caused by biofilm-forming bacteria.

## 1. Introduction

Honey is a natural substance produced by *Apis mellifera* bees from the nectar of plants or from secretions of living parts of plants or excretions of plant-sucking insects on the living parts of plants, which the bees collect and transform by combining with specific substances of their own [1]. Honey of all origins is composed mainly of the sugars glucose, fructose, and sucrose, which constitute ∼80% of its weight, with water composing the remaining 20%. In addition, vitamins, flavonoids, amino acids, enzymes, minerals, and phenolic acids are also present in honey [2]. Honey is one of the oldest traditional medicines, often applied as a remedy for microbial infections [3]. Honey can prevent and relieve the symptoms of respiratory tract infections, as well as gastrointestinal and cardiovascular diseases, due to the antioxidant, anti-inflammatory, antiviral, and antibacterial effects [4,5,6,7,8,9,10,11,12]. Both Gram-positive and Gram-negative pathogenic bacteria are susceptible to honey, including methicillin-resistant *Staphylococcus aureus* (MRSA) [13], *Shigella sonnei* [14], *Helicobacter pylori* [15], and yeasts like *Candida albicans* [16].

Recently, antibacterial resistance poses a major problem, which is in correlation with the biofilm forming ability of bacteria. Many of the resistant bacterial cells take advantage of being embedded in a biofilm, which is a complex matrix of polysaccharides and other components, providing optimal conditions and protection for the bacterial community [17,18,19,20]. Infections associated with biofilm growth are usually challenging to eradicate, since mature biofilms display tolerance towards environmental stress factors, as well as antibiotics and the immune response. Biofilms are compact structures of bacteria, which can resist the worst environmental circumstances and even survive the antibiotic treatment. Furthermore, during the antibiotics treatment the subclinical concentration can diffuse into the biofilm, which is not enough to kill the bacterial cells, but it can enhance the appearance of antibiotics resistance genes [21,22]. Moreover, biofilms can cause chronic diseases, as well [23,24]. The respiratory tract is the barrier between the environment and the human body, which may facilitate pathogens entering this area. The most common respiratory tract bacteria are *Haemophilus* spp., *Pseudomonas aeruginosa*, and *Streptococcus pneumoniae* [25,26]. *Haemophilus* species with *Aggregatibacter* species, *Cardiobacterium hominis*, *Eikenella corrodens*, and *Kingella* species (HACEK group) are a small, heterogeneous group of fastidious, Gram-negative bacteria that frequently colonize the oropharynx, and can cause endocarditis due to their biofilm-forming ability. In the respiratory tract, they are often responsible for chronic otitis media, pharyngitis, and laryngitis [27,28,29]. *P. aeruginosa* is a well-known pathogen, which can develop biofilms on respiratory tract mucosa, being the causing agent of otitis media, sinusitis, and pharyngitis. This is the most significant Gram-negative bacterium in nosocomial diseases [30,31,32]. *P*. *aeruginosa*’s biofilm is highly resistant to antibiotics and disinfectants, due to its ability to produce alginate mucus [17,33]. Besides causing pneumonia, *S. pneumoniae* is common in otitis media and sinusitis, sepsis, and meningitis as well [34,35,36]. This Gram-positive bacterium can stick to the surface of bronchi easily, where it can create biofilms, making treatment more difficult [37,38]. The significance of the above-mentioned pathogens lies in the high risk they pose in health care. Due to the increased frequency of antibiotic resistance and the biofilm forming ability of several bacteria, it is essential to research new and natural antibacterial and antibiofilm agents, including honeys of various botanical origin. 

There are data on the physicochemical properties, as well as sugar and volatile composition of black locust, linden, and sunflower honeys produced in various countries [39,40,41,42], including Hungary [43,44,45], but data are lacking about the antibacterial activity of Hungarian honeys. Our recent study, regarding the antioxidant activity of eight different unifloral honeys from Hungary (acacia/black locust, amorpha, phacelia, linden, sunflower, chestnut, fennel and meadow sage), revealed that linden honey had exceptionally high antioxidant activity despite its light colour [46]. Our hypothesis was that linden honey will be effective also as an antibacterial agent. From the eight Hungarian honey types mentioned above, for the purposes of the present study, we chose the ones that are readily available for customers and have different levels of antioxidant capacity: black locust with pale colour and low antioxidant activity, linden honey with light colour and high antioxidant activity, and sunflower honey with darker colour and high antioxidant activity. The aim of this study was to evaluate the antibacterial and antibiofilm properties of Hungarian black locust, linden, and sunflower honey samples against both Gram-negative and Gram-positive respiratory tract bacteria in three in vitro test systems. In addition, we intended to gain insight into the kinetics and possible mechanisms of action of the antibacterial activity, therefore time-kill assays and membrane degradation studies were performed as well.

## 2. Materials and Methods

### 2.1. Melissopalynological Analysis

The honey samples were purchased in 2020 from two Hungarian apiaries, directly from beekeepers, who identified the samples as black locust, linden, and sunflower honeys. The botanical origin of each honey sample was checked with melissopalynological analysis [47]. Honey samples, when fluid, were stirred thoroughly. In case they contained large crystals, they were heated on a 40 °C water bath, until fluid, then stirred. 10 g of honey was measured into 50-mL centrifuge tubes, 20 mL of distilled water was added, then vortexed with Combi-spin FVL-2400N (Biocenter Ltd., Szeged, Hungary). The solution was centrifuged at 3000 rpm for 10 min with a Neofuge 15R centrifuge (Lab-Ex Ltd., Budapest, Hungary). The supernatant was decanted, then 10 mL of distilled water was added to the sediment, and this mixture was centrifuged again at 3000 rpm for 10 min, and decanted. Any remaining fluid was removed by setting the centrifuge tubes on filter paper. A frame of the size of the cover glass was drawn on each microscope slide with a paint marker (Edding 750), then the microscope slides were placed on a heating plate (OTS 40, Tiba Ltd., Győr, Hungary) set to 40 °C. Then, 0.25 mL of distilled water was added to the sediment in the centrifuge tube, then vortexed. A volume of 20 μL of the pollen suspension was pipetted on the microscope slide within the frame. Water was allowed to evaporate from the slide on the heating plate. The pollen preparation was mounted in fuchsine glycerol jelly (fuchsine added to Kaiser’s glycerol jelly). Pollen preparations were studied with a Nikon Eclipse E200 microscope equipped with a Michrome 20MP CMOS digital camera (Auro-Science Consulting Ltd., Budapest, Hungary), and microphotos were taken with the software Capture 1.2 at 400× magnification. At least 500 pollen grains per honey sample were counted, and the source plants were identified at species level, or at least at family level. The relative frequency for each type of pollen was calculated as the percentage of the total number of pollen grains.

### 2.2. Thin Layer Chromatography–Direct Bioautography (TLC–DB)

The antibacterial effect of honey samples was screened against *Haemophilus influenzae* (DSM 4690), *H. parainfluenzae* (DSM 8978), *Pseudomonas aeruginosa* (PAO I), and *Streptococcus pneumoniae* (DSM 20566). For growing bacteria, Brain Heart Infusion Broth (BHI) (Sigma Aldrich Ltd., Darmstadt, Germany) was used, and in the case of *Haemophilus* species, 1 mL of supplement B (Diagon Kft., Budapest, Hungary) and 15 µg/mL NAD solution (1 mg/mL) was added to BHI. After that, the solution was shaken at 37 °C in a shaker incubator (C25 Incubator Shaker, New Brunswick Scientific, Edison, NJ, USA) at a speed of 60 rpm for 24 h. The honey samples were dissolved in sterilized distilled water. The concentration of the stock solution was 1 g/mL in each honey sample. From these solutions 1.0 μL was applied to the silica gel 60 F254 aluminum sheet 5 × 10 cm TLC plates (Merck, Darmstadt, Germany) with Thermo Fisher Scientific Finnpipette (Merck, Darmstadt, Germany), so the substance content on the plate was 1 mg. The negative control was the distilled water, the positive control was gentamicin (Sandoz, 40 mg/mL) against *P. aeruginosa*, ceftriaxone (Hospira, 250 mg powder, stock solution: 40 mg/mL) against *Haemophilus* spp., and *S. pneumoniae*. From the antibiotic samples we applied 1.0 μL. The plates were dipped into 100 mL of bacterial suspensions (4 × 10^7^ CFU/mL). After that the plates were put in low-wall horizontal chambers, with dimensions of 20 × 14.5 × 5 cm. The incubation time was 1.5 h and the temperature of the box was 37 °C. In order to visualize the inhibition zones, MTT 3-(4,5-dimethylthiazol-2-yl)-2,5-diphenyltetrazolium bromide, Sigma Aldrich Ltd.) solution (0.05 g MTT powder dissolved in 90 mL of water) was used. The MTT solution is a bacterial selective dye, the color of which will change to blue or violet in the presence of viable bacteria. After using MTT solution, the incubation time was 12 h (37 °C). The antibacterial activity of the samples, following 13.5 h of total incubation time, was shown by the white inhibitory zones on the plates. The diameter of inhibition zones (expressed in mm) was measured with Motic Images Plus 2.0 program (Motic Deutschland GmbH, Wetzlar, Germany). All tests were carried out in eight replicates.

Since our preliminary statistical analysis revealed no significant differences between honey samples from different apiaries, but with the same botanical origin, in the subsequent microbiological assays we worked with pooled black locust, linden, and sunflower honey samples.

### 2.3. Agar Well Diffusion Assay

In this assay, the antibacterial activity of honey samples and sugar solutions (glucose:fructose, 1:1) was examined. We used chocolate agar for *Haemophilus* species and *Streptococcus pneumoniae*, and Müller-Hinton agar for *Pseudomonas aeruginosa*. In order to compare the antibacterial effects of honeys and sugar solutions, honey samples and also glucose and fructose (Sigma Aldrich Ltd.) were diluted in sterilized distilled water to 50% (*w/w*) and 25% (*w/w*). As a positive control, we used gentamicin against *P. aeruginosa*, ceftriaxone against *Haemophilus* spp., and *S. pneumoniae*. The stock solutions from antibiotics were 5 µg/mL. We added 150 µL from honey and sugar solutions, and 75 µL from antibiotic solutions to the wells. The plates were incubated at 37 °C for 12 h, after which period the diameter of the inhibition zones (expressed in cm) was measured. All tests were carried out six times.

### 2.4. Time-Kill Assay

Time-kill studies were conducted with escalating concentrations of honey samples and glucose:fructose (1:1) solution against *Pseudomonas aeruginosa* (Gram−) and *Streptococcus pneumoniae* (Gram+). The bacterial suspensions were diluted with BHI (37 °C, 12 h); 100 µL of the suspension was transferred to 96-well microtiter plates, each containing 100 µL of a honey or sugar solution at 20%, 40%, and 60% (*w*/*w*) concentrations. The final concentrations of the bacterial suspension in each well at baseline were approximately 10^5^ CFU/mL. As a control, we used the bacterial suspensions without treatment. We measured the absorbance (600 nm) seven times (0, 2, 4, 6, 8, 12, and 24 h) with plate reader (BMG Labtech, SPECTROstar Nano).

### 2.5. Membrane Degradation Study 

The release of cellular material was examined in *P. aeruginosa* (Gram−) *and S. pneumoniae* (Gram+). The absorbance of 1 mL bacterial suspension containing 2 × 10^7^ CFU/mL in PBS (phosphate buffer saline) was measured at 260 nm. The bacterial cells treated with honey were suspended for 1 h in PBS containing 20%, 40%, and 60% (*w*/*w*) concentrations of honey samples. Control cells were suspended in PBS without honey treatment. As positive control 90% solution of honey samples was used. In order to examine the antibacterial effect of sugar components, glucose:fructose (1:1) solutions were tested, as well. Moreover, the bacterial cells suspended in PBS containing honey and sugar samples (60%) were treated for different periods of time: 0, 10, 20, 40, 60, and 90 min. After treatment, cells were centrifuged (Neofuge 15R, Lab-Ex Ltd., Budapest, Hungary) at 11,107 rpm (12,000× g) for 2 min, and the absorbance of the supernatant at 260 nm was determined in Metertech SP-8001 (Abl&E-Jasco Ltd., Budapest, Hungary) spectrophotometer [48,49]. The results were expressed in percentage values, which were compared to the untreated cells. 

### 2.6. Microdilution Assay

During biofilm inhibition experiments, honey samples with half the minimum inhibitory concentration (MIC/2) were used. The MICs were determined with broth microdilution test. We used 96-well microtiter plates to perform this assay. From each bacterium solution (10^5^ CFU/mL) 100 μL was measured to the wells. From our honey samples 12.5%, 25%, 40% 50%, and 70% (*w*/*w*) stock solutions were prepared. Our preliminary experiments revealed that the MIC value will be between 40–55%, thus a dilution series of 40%, 40.5%, 41%, 41.5% to 55.5% was made. From stock solutions, 100 μL was added to each well. After incubation (24 h, 37 °C) we measured the absorbance (600 nm) with plate reader (BMG Labtech, SPECTROstar Nano). Distilled water was used as negative control, and the untreated bacterial suspension as positive control. All tests were carried out in six replicates, from which the average value was calculated, and then the mean of the negative control was subtracted from the value obtained. The concentration at which the absorbance was lower than 10% of the positive control samples, i.e. bacterial growth was inhibited by 90% or more, was considered as the MIC value.

### 2.7. Antibiofilm Activity 

The biofilms were prepared in 96-well microtiter plates. 200 µL of bacterial culture (4 × 10^7^ cells/mL) was added into each well, then the microtiter plate was incubated at 37 °C for 4 h in order to help the adhesion of the cells. After the incubation time the non-adherent cells were washed with physiological saline solution. Distilled water was used as a negative control, and the untreated bacterial suspension as a positive control. The biofilms were treated with 20% solution of honey samples. After the treatments the microtiter plate was incubated at 37 °C for 24 h. Then the adherent cells were fixed with methanol for 20 min. The biofilms were dyed with 0.1% crystal violet solution for 25 min. 33 *w*/*w*% of acetic acid was added to each well, then the absorbance intensity was measured at λ= 595 nm with microtiter plate reader (BMG Labtech SPECTROstar Nano). All tests were carried out six times [50].

### 2.8. Statistical Analysis

The data were compared with One-way ANOVA with Tukey’s pairwise comparisons. If the normality assumption was violated, we applied Kruskal–Wallis test with Mann–Whitney pairwise comparisons. Differences were considered statistically significant at *p* ≤ 0.05. All statistical data were calculated using Past statistic software (Version 2.17b; [51]).

## 3. Results

### 3.1. Melissopalynological Analysis

The botanical origin of the honey samples, identified as black locust (*Robinia pseudoacacia)*, linden (*Tilia* sp.), and sunflower (*Helianthus annuus*) honeys by the producers, was confirmed by means of microscopic pollen analysis. The relative frequencies of pollen types in each honey sample are summarized in Table 1. In black locust honey samples, the dominant pollen was *R. pseudoacacia* (Figure 1a), accompanied by pollen grains of *Brassica napus*, *Solidago* sp., and *Tilia* sp. (Figure 1b). In linden honeys, the majority of pollen grains belonged to *Tilia* sp., but *H. annuus* and *R. pseudoacacia* pollen was also identified in lower percentage. Sunflower honeys contained *H. annuus* pollen (Figure 1c) as their dominant pollen type.

### 3.2. Antibacterial Activity of Honeys

#### 3.2.1. Thin Layer Chromatography–Direct Bioautography (TLC–DB) Assay

Thin layer chromatography–direct bioautography (TLC–DB) is a well-known method to detect the antibacterial activity of dissolved samples, but it is less frequently used with honey samples [52]. Thus, as the first step, we optimized the TLC–DB assay successfully, which allowed us to detect the inhibition zones of honey samples against respiratory tract pathogens as shown for *P. aeruginosa* in Figure 2.

Distilled water as a negative control did not inhibit the growth of any of the bacteria, while the 1-µL solution of the antibiotics (gentamicin against *P. aeruginosa*, ceftriaxone against *Haemophilus* spp. and *S. pneumoniae*) was effective against each bacterial strain. Our results showed that all honey samples were active against each bacterium. Linden honey was significantly more active against *Haemophilus* and *Pseudomonas* strains, compared to black locust honey samples, while the activity of sunflower honey did not differ statistically from that of either black locust or linden honey. In the case of *S. pneumoniae*, both black locust and linden honeys were significantly more active compared to sunflower honey, but the antibacterial effect of black locust and linden honeys did not differ statistically from each other (Appendix A).

Comparing the two *Haemophilus* species, *H. parainfluenzae* was more sensitive to each honey sample than *H. influenzae*, indicated by larger zones of inhibition, with average values of 4.0 and 3.8 mm for linden honey, and 3.5 and 3.2 mm for black locust honey, respectively. Treatment with linden honey resulted in larger inhibition zones, compared to black locust honey, both against *P. aeruginosa* (4.1 and 3.5 mm, respectively), and *S. pneumoniae* (3.9 and 3.6 mm, respectively). Sunflower honey inhibited each bacterium strain to a similar degree (Figure 3).

#### 3.2.2. Agar Well Diffusion Assay

As Table 2 and Table 3 show, each honey sample proved to be more effective compared to sugar solutions, although the latter substances displayed some antibacterial activity, too. However, in the agar well diffusion assay, no significant differences could be detected in the antibacterial activity of various honey types. Regarding the sensitivity of bacterial strains, the lowest antibacterial effect was detected in case of *P. aeruginosa*, whose growth was inhibited to lesser degree by honey treatments, compared to other bacteria (Appendix A).

### 3.3. Kinetics and Mechanisms of Action of Antibacterial Activity

#### 3.3.1. Time-Kill Assay

The time-kill assay kinetics additionally confirmed the bactericidal potential of 40% (*w*/*w*) and 60% (*w*/*w*) black locust, linden, and sunflower honey samples against *S. pneumoniae* (Gram+) and *P. aeruginosa* (Gram−) bacteria. As a control, we used sugar solutions (glucose:fructose; 1:1) in order to exclude the sole antibacterial activity of the sugar content in honey. Our results showed that each honey sample exhibited antibacterial activity from the second hour, compared to the control. Between 6 and 8 hours after treatment with honeys and sugar controls, the growth of *P. aeruginosa* and *S. pneumoniae* was retained steadily. We can conclude that some antibacterial activity can be attributed to the sugar samples, however, they showed lower activity than honey samples against both bacterial strains. Linden honey was the most effective sample against both bacterial strains compared to black locust and sunflower samples, as shown by Figure 4 in the case of *P. aeruginosa*.

#### 3.3.2. Membrane Degradation Study

In order to investigate potential mechanisms of action of honey, we studied the process of bacterial membrane degradation, by measuring the degree of bacteriolysis in *P. aeruginosa* and *S. pneumoniae* cells, treated with a range of concentrations of sugar and honeys. The 20% solutions of sugar and honey did not cause any lysis; however, the 40% and 60% solutions were active. *S. pneumoniae* responded more sensitively to treatment than *P. aeruginosa*. Each honey sample performed better than the sugar solutions, and from the honey samples, linden honey showed the best activity against both bacteria (Table 4).

In order to investigate the kinetics of the release of cellular material, we performed time course lysis with 60% (*w*/*w*) honey and sugar solutions, assessing the degree of cell lysis after various time intervals, up to 90 min (Table 5). Contact for 20 m with these antibacterial agents was shown to be enough to induce consistent release of 260-nm absorbing material. Our results showed that sugar solutions had lower activity compared to honey samples. From honey samples, linden honey was the most effective in case of both bacteria. The highest degree of membrane degradation was detected at 90 min, following treatment with linden honey.

Based on the above experiments we can conclude that black locust, linden, and sunflower honey solutions can induce bacteriolysis, by disrupting the membrane of bacterial cells, not only in case of Gram−, but also in Gram+ bacterial strains. It was also proven that the sugar components of honey can be responsible for some of this activity, nevertheless, honey samples had higher membrane degradation activity compared to sugars alone in each experimental setup.

### 3.4. Antibiofilm Activity

#### 3.4.1. Broth Microdilution Test

The MIC value of honey samples was determined by microdiluton assay. Our samples showed similar activity, their MIC value ranging between 40.5 and 52.5% (Table 6). The most sensitive strains were the *Haemophilus* spp., their growth being inhibited by 40.5% concentration of linden honey. The most resistant pathogen was *P. aeruginosa*, inhibited only by the 50.5% solution of linden honey, and the 52.5% solution of black locust and sunflower honeys. Afterwards, a two-fold dilution of the MIC value (MIC/2 value) was used in the antibiofilm assay.

#### 3.4.2. Antibiofilm Activity

The potential of each honey sample to inhibit the biofilm forming ability of various bacterial strains was examined with MIC/2 concentration. The anti-biofilm formation activity of the honey samples was calculated and demonstrated in terms of inhibitory rate [53] (Figure 5). Our experiments revealed that each honey sample in this study was able to inhibit the formation of biofilm by each bacterium strain tested. Similar to the results of the TLC–DB assays, linden honey showed the best performance also regarding the inhibition of biofilm formation, with mean inhibitory rates above 80% against each strain (80 to 82%). In contrast, black locust samples exerted the lowest inhibition against respiratory tract biofilms, with inhibitory rates ranging between 69 and 75% (Appendix A). It should be highlighted that *S. pneumoniae* was the most sensitive bacterium from the respiratory tract pathogens (Figure 5).

## 4. Discussion

The antibacterial activity of honey is highly complex and still remains not fully recognized. To date, it has been established that the mechanism of antibacterial effect varies, attributed to the fact that honey can have multiple attack points. The high concentration of sugars (80%) can inhibit the growth of bacteria, due to the high osmotic pressure. Moreover, the Gram-positive bacteria (like *S. pneumoniae*) are sensitive to defensin-1, which is a peptide secreted by the honeybee hypopharyngeal glands. One of the most important enzymes involved in honey ripening processes is glucose-oxidase, which can transform glucose to gluconic acid. The side product of this reaction, hydrogen peroxide is a strong antibacterial agent [12,54,55,56,57,58]. Besides, gluconic acid itself causes low pH-values (3.4–6.1) in honey, which prevents the development of most microorganisms. However, the antimicrobial efficiency of each honey type is not only dependent on its glucose oxidase activity, but also on its profile and concentration of honey constituents, particularly phenolic compounds [59,60,61]. 

Many bacteria were reported to display a high sensitivity to honey, particularly *Staphylococcus aureus*, *Streptococcus pneumoniae*, *Streptococcus pyogenes*, *Helicobacter pylori*, *Mycobacterium tuberculosus*, *Escherichia coli*, *Psuedomonas aeruginosa*, *Micrococcus luteus*, and *Bacillus subtilis* [12,58,62,63,64], but there were no previous studies that focused exclusively on the reaction of respiratory tract bacteria following treatment with honey. For this reason, we chose to test the antibacterial potential and anti-biofilm activity of black locust, linden, and sunflower honeys, three well-known honeys that are readily available in Hungary, against the most common respiratory tract bacteria, including *H. influenzae* (DSM 4690), *H. parainfluenzae* (DSM 8978), *P. aeruginosa* (PAO I.), and *S. pneumoniae* (ATCC 20566), using agar well diffusion, time-kill and biofilm inhibitory assays, as well as TLC–DB. Since the latter technique was used only in a few instances with honey, e.g. [52], our research group had to optimize the process for use with honey samples. Optimizing TLC–DB for two *Haemophilus* species involved altering conditions such as incubation time, and the composition of agar for growing the bacterium, the necessity of which was also confirmed in our previous study [65]. 

Brown and co-workers proved the anti-*Haemophilus* activity of various honey samples (produced by *Apis mellifera, Frieseomelitta nigra*, and *Melipona favosa* bees), using agar diffusion assay [66]. Al-Waili’s research group examined the antibacterial activity of multifloral honey against *Staphylococcus aureus*, *S. hemolyticus*, *Escherichia coli*, *Klebsiella* sp., *P. aeruginosa*, *Enterobacter cloacae*, *H. influenzae*, and *Proteus* sp., because these are the most common bacteria in wounds. The results of this research group showed that the growth of *S. aureus*, *H. influenzae,* and *P. aeruginosa* was inhibited by the 30–50% multifloral honey samples, which values were comparable to the MIC values obtained in our study, being 40.5–42% and 50.5–52.5% for *H. influenzae* and *P. aeruginosa*, respectively [67]. Newby et al. described that *H. influenzae* biofilm was sensitive to SurgihoneyRO (a licensed, CE-marked sterile honey that has been bioengineered to enable the controlled release of H_2_O_2_ over a prolonged period), because this honey was able to cause significant reduction in biofilm growth [68]. It has to be highlighted that the bioactivity of honey against *H. parainfluenzae* has not been described yet. Our study was the first to prove that black locust, linden, and sunflower honeys were active against both *H. influenzae* and *H. parainfluenzae*, the highest activity shown by linden honey both in TLC–DB and antibiofilm assays. 

Salonen et al. compared the antibacterial effect of sweet clover (*Melilotus officinalis*), buckwheat (*Fagopyrum esculentum*), multifloral, raspberry (*Rubus idaeus*), and heather (*Calluna vulgaris*) honeys, as well as honeydew against *P. aeruginosa* by microdilution assay. Their experiments revealed that from the five honey samples, buckwheat and raspberry honey inhibited the growth of *P. aeuruginosa*, similarly to the honeydew sample [69]. Our findings are in accordance with Awan and co-workers’ results regarding the anti-*Pseudomonas* activity of sunflower honey, which was, however, inferior compared to the effect of eucalyptus honey [70]. Manuka (*Leptospermum scoparium*) honey proved to be effective against *P. aeruginosa* both in antibacterial and antibiofilm assays [71,72,73]. In addition, clover (*Trifolium* sp.), buckwheat and sidr (*Ziziphus spina-christi*) honey samples were found to be effective against *P. aeruginosa* biofilm [74]. A recent study supported the antibacterial activity of lime (linden), rapeseed (*Brassica napus*), buckwheat, and multifloral honey samples against *P. aeruginosa* [75]. In the latter study, linden honey was one of the most effective honey samples against this bacterium, similarly to our findings. Another recent study revealed that multiple mechanisms of actions are responsible for the antibacterial activity of pine honey against *P. aeruginosa*, exerting an inhibitory effect on quorum sensing, bacterial chemotaxis and biofilm formation [76].

The reaction of *S. pneumoniae* to honey treatment has received very limited attention so far. Some research supported that *Streptococcus* spp. in general were sensitive to treatment with honey [77,78,79]. From five different types of Finnish honeys, willow herb (*Epilobium angustifolium*) honey was found to be the most effective against *S. pneumoniae*, compared to heather, lingonberry (*Vaccinium vitis-idaea*), buckwheat, and cloudberry (*Rubus chamaemorus*) honey samples. Huttunen highlighted that *S. pneumoniae* was the most sensitive bacterium to the treatment (microbroth dilution assay), in comparison with *S. pyogenes* and *Staphylococcus aureus* [80]. Our study was the first to present data regarding the effect of black locust, linden, and sunflower honeys on the growth and biofilm formation ability of *S. pneumoniae*. Similarly to the Finnish honeys, we found that honey concentrations slightly above 40% were necessary to inhibit the growth of this bacterium, when treated with the three Hungarian honeys involved in our study (MIC values in the range 42.5–45%).

As for the kinetics of the antibacterial activity, time-kill studies revealed that our honey samples started inhibiting bacterial growth between 6 and 8 h of incubation both for *P. aeruginosa* and *S. pneumoniae*, similarly to a recent study performed with Iranian honeys and *P. aeruginosa* [73].

An important component of honey’s antimicrobial activity is the disruption of bacterial membrane. Due to membrane damage, honey can induce bacteriolysis, both in the case of Gram-negative and Gram-positive bacteria [49]. In our study, *S. pneumoniae* (Gram+) was more sensitive compared to *P. aeruginosa* (Gram−) both in the time-kill assay and membrane degradation studies, which is likely due to structural differences. Similarly, Otmani et al. [81] found that a Gram+ strain (*Staphylococcus aureus* FRI S6) was inhibited to larger extent by honey samples, compared to a Gram- strain (*Salmonella typhi* ATCC 14028).

## 5. Conclusions

This study highlighted the effectiveness of commonly available Hungarian honey types against the essential respiratory tract bacteria. To the best of our knowledge, our research group was the first to demonstrate the bioactivity of honey against *H. parainfluenzae*, as well as the inhibitory effect of black locust, linden, and sunflower honeys on *S. pneumoniae*; thus, the novelty of our study is indisputable. Our findings support the view that honeys of various botanical origin can provide an alternative means to treat respiratory tract infections, due to their potential to decrease the growth and biofilm production and disrupt the membrane of respiratory tract bacteria. At the same time, our study revealed that even in the case of honeys from the same geographical region, the botanical source of the honey can have a high impact on its biological activity. From the investigated Hungarian honey types, linden honey particularly deserves more attention as a powerful natural antibacterial agent.

## Figures and Tables

**Figure 1 foods-10-01632-f001:**
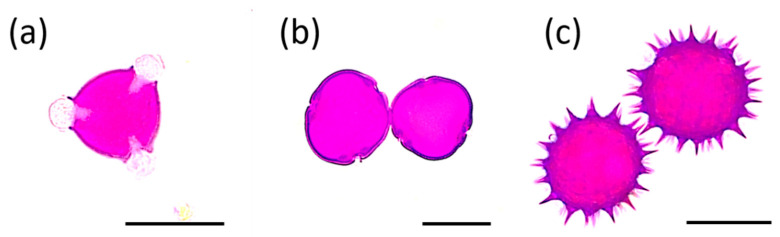
Analysis of botanical origin of honey samples with light microscopy. Pollen grains of: (**a**) black locust, (**b**) linden, and (**c**) sunflower. Scale bar = 25 μm.

**Figure 2 foods-10-01632-f002:**
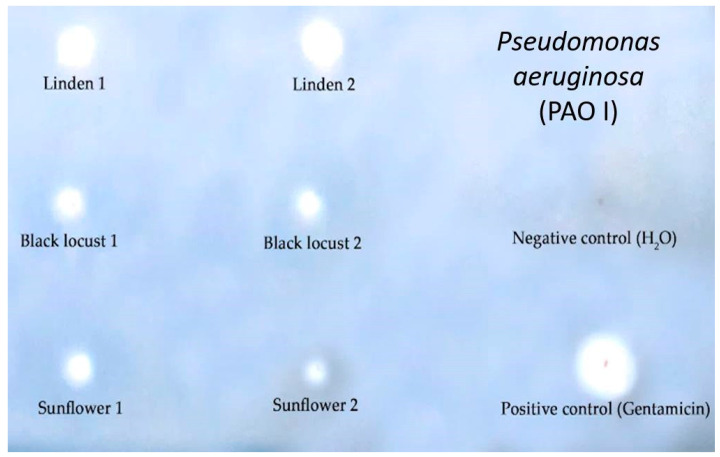
TLC–DB plate in the case of *P. aeuruginosa*. Negative control: distilled water, Positive control: gentamicin (40 mg/mL) from which 1.0 μL was applied to the plate. The numbers after each honey type refer to Apiary 1 and Apiary 2. The stock solutions from honey samples were 1.0 g/mL, 1.0 μL was applied to the plate. The blue background shows the viable bacteria. The white spots (as inhibition zones) indicated the lack of dehydrogenase activity due to the antibacterial activity of honey samples.

**Figure 3 foods-10-01632-f003:**
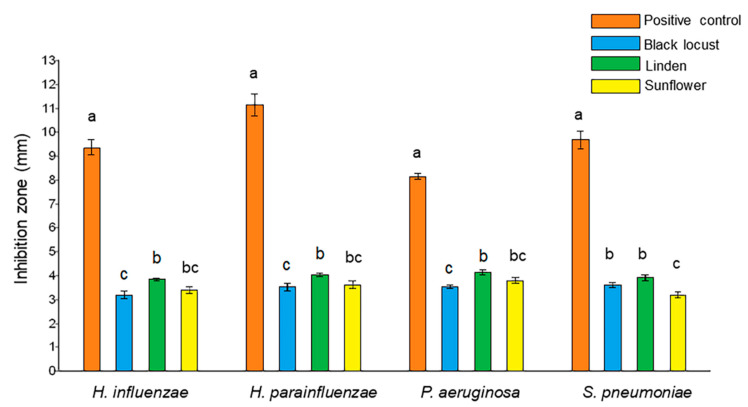
Antibacterial activity of black locust, linden, and sunflower honey samples against respiratory tract bacteria. The diameter of the inhibition zones was expressed in mm. Positive controls: gentamicin against *P. aeruginosa*, ceftriaxone against *Haemophilus* spp., and *S. pneumoniae*. Different letters above bars representing the same bacterial strain indicate significant differences at *p* ≤ 0.05.

**Figure 4 foods-10-01632-f004:**
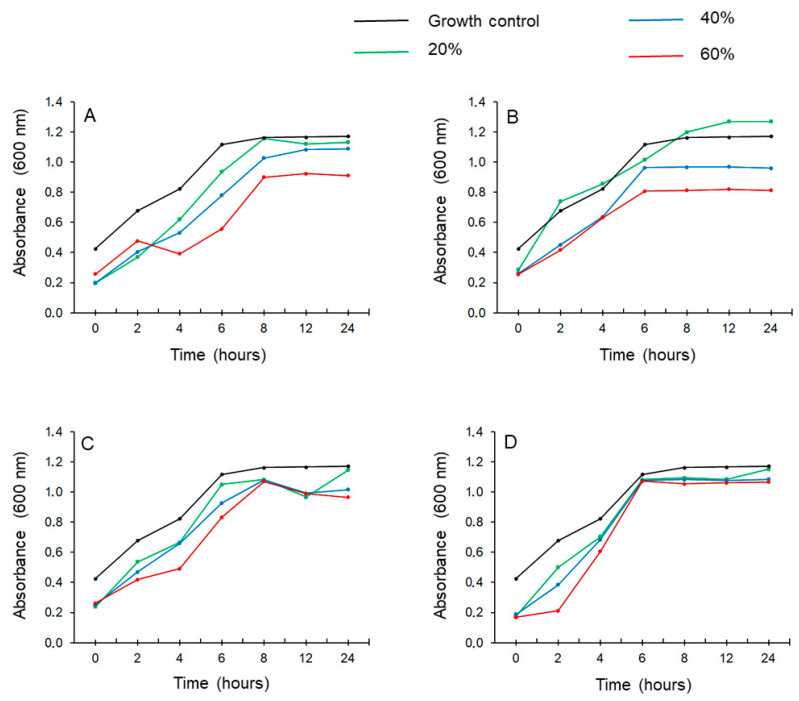
Time-kill assay in case of *P. aeruginosa:* (**A**) black locust honey samples, (**B**) linden honey samples, (**C**) sunflower honey samples, and (**D**) sugar (glucose:fructose, 1:1) solutions.

**Figure 5 foods-10-01632-f005:**
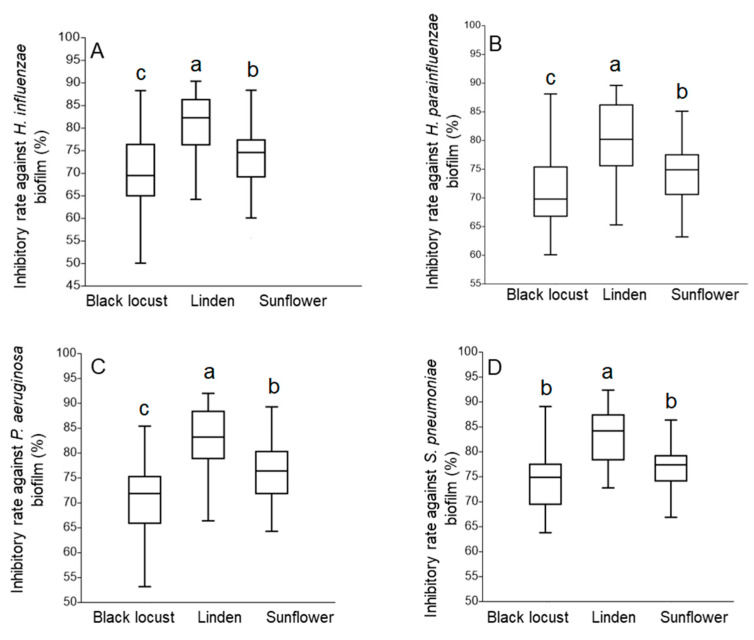
Biofilm inhibition activity of different honey samples against respiratory tract bacteria: (**A**) *Haemophilus influenzae*, (**B**) *H. parainfluenzae*, (**C**) *Pseudomonas aeruginosa*, and (**D**) *Streptococcus pneumoniae*. The anti-biofilm formation activity was calculated and demonstrated in terms of inhibitory rate according to the equation: Inhibitory rate = (1 − S/C) × 100% (C and S were defined as the average absorbance of control and sample groups, respectively). Different lower case letters (a, b, c) above boxes in figures (**A**–**D**) indicate significant differences at *p* ≤ 0.05.

**Table 1 foods-10-01632-t001:** Relative frequency of pollen types in Hungarian black locust, linden, and sunflower honeys.

Honey Type	Pollen Type—Relative Frequency (%)						
	*Robinia*	*Tilia*	*Helianthus*	*Solidago*	*Brassica*	Apiaceae	Other
black locust *R. pseudoacacia*	67.0	0.5	-	13.8	5.4	0.5	12.8
linden *Tilia* sp.	27.6	41.0	16.1	-	-	-	15.3
sunflower *H. annuus*	16.3	7.4	48.1	3.8	-	2.3	22.1

**Table 2 foods-10-01632-t002:** Results of the agar well diffusion assay with 25% honey and sugar solutions, compared to antibiotics as positive control. Inhibition of bacterial growth was indicated by the appearance of a zone of inhibition around the well.

	Diameter of Inhibition Zones (cm) ^1^			
Test Solution	*H. influenzae*	*H. parainfluenzae*	*P. aeruginosa*	*S. pneumoniae*
black locust 25%	1.7 ± 0.2 ^a^	1.6 ± 0.3 ^a^	1.2 ± 0.1 ^ab^	1.6 ± 0.2 ^ab^
linden 25%	1.8 ± 0.2 ^a^	1.8 ± 0.2 ^a^	1.4 ± 0.2 ^a^	1.7 ± 0.2 ^a^
sunflower 25%	1.5 ± 0.1 ^ab^	1.6 ± 0.2 ^a^	1.0 ± 0.2 ^b^	1.3 ± 0.2 ^b^
sugar solution 25%	1.2 ± 0.2 ^b^	1.1 ± 0.1 ^b^	0.9 ± 0.2 ^b^	0.9 ± 0.3 ^c^
antibiotic control	3.5 ± 0.3 ^c^	3.5 ± 0.3 ^c^	3.4 ± 0.2 ^c^	3.6 ± 0.3 ^d^

^1^ Data are presented as means ± standard deviations, based on 6 parallel measurements. Different superscript letters within the same column indicate significant differences at *p* ≤ 0.05.

**Table 3 foods-10-01632-t003:** Results of the agar well diffusion assay with 50% honey and sugar solutions, compared to antibiotics as positive control. Inhibition of bacterial growth was indicated by the appearance of a zone of inhibition around the well.

	Diameter of Inhibition Zones (cm) ^1^			
Test Solution	*H. influenzae*	*H. parainfluenzae*	*P. aeruginosa*	*S. pneumoniae*
black locust 50%	1.9 ± 0.3 ^a^	1.9 ± 0.2 ^a^	1.7 ± 0.3 ^a^	1.9 ± 0.2 ^a^
linden 50%	2.1 ± 0.1 ^a^	2.1 ± 0.1 ^a^	1.9 ± 0.3 ^a^	2.0 ± 0.1 ^a^
sunflower 50%	1.8 ± 0.2 ^a^	1.8 ± 0.2 ^a^	1.7 ± 0.2 ^a^	1.9 ± 0.2 ^a^
sugar solution 50%	1.4 ± 0.2 ^b^	1.3 ± 0.3 ^b^	1.2 ± 0.1 ^b^	1.3 ± 0.1 ^b^
antibiotic control	3.5 ± 0.3 ^c^	3.5 ± 0.3 ^c^	3.4 ± 0.2 ^c^	3.6 ± 0.3 ^c^

^1^ Data are presented as means ± standard deviations, based on 6 parallel measurements. Different superscript letters within the same column indicate significant differences at *p* ≤ 0.05.

**Table 4 foods-10-01632-t004:** The effect of sugar and honey solutions at different concentrations on the release of cellular material, absorbing at 260 nm, from *P. aeruginosa* and *S. pneumoniae*. The degree of bacteriolysis was expressed in percentage compared to the control.

Concentrations (%)	Lysis of *P. aeruginosa* Cells				Lysis of *S. pneumoniae* Cells			
	Sugar	Black Locust	Linden	Sunflower	Sugar	Black Locust	Linden	Sunflower
	A_260_ (%)				A_260_ (%)			
0	0	0	0	0	0	0	0	0
20	0	0	0	0	0	0	0	0
40	22.1	25.8	32.9	27.0	26.4	31.6	39.5	30.9
60	30.2	36.8	40.2	35.9	33.7	38.9	42.5	40.1
90	100	100	100	100	100	100	100	100

**Table 5 foods-10-01632-t005:** Kinetics of 260-nm absorbing material release from *P. aeruginosa* and *S. pneumoniae* treated with 60% (*w/w*) sugar and honey solutions.

Time (min)	Lysis of *P. aeruginosa* Cells				Lysis of *S. pneumoniae* Cells			
	Sugar	Black Locust	Linden	Sunflower	Sugar	Black Locust	Linden	Sunflower
	A_260_ (%)				A_260_ (%)			
0	0	0	0	0	0	0	0	0
10	0	0	0	0	0	0	0	0
20	15.8	22.4	26.1	21.9	17.1	25.6	30.1	24.8
40	23.5	32.5	36.2	33.1	24.1	34.5	38.6	34.7
60	30.2	36.8	40.2	35.9	33.7	38.9	42.5	40.1
90	35.8	45.2	51.6	46.1	37.1	47.2	54.9	50.6

**Table 6 foods-10-01632-t006:** The MIC and MIC/2 value of linden, black locust, and sunflower honey.

	Honey Samples	1	2	3	4
	black locust	42.0%	42.0%	52.5%	44.0%
MIC value	linden	40.5%	40.5%	50.5%	42.5%
	sunflower	42.0%	42.0%	52.5%	45.0%
	black locust	21.0%	21.0%	26.3%	22.0%
MIC/2 value	linden	20.3%	20.3%	25.3%	21.3%
	sunflower	21.0%	21.0%	26.3%	22.5%

1: *H. influenzae*, 2: *H. parainfluenzae*, 3: *P. aeruginosa*, 4: *S. pneumoniae.* Percentage values of the table correspond to dilution % of honey causing antimicrobial effect.

## Data Availability

Data are contained within the article and Appendix A.

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
