# Peer review of "In Vitro Antibacterial and Antibiofilm Activity of Hungarian Honeys against Respiratory Tract Bacteria"

_foods, 2021, doi:10.3390/foods10071632_

Round 1

Reviewer 1 Report

The manuscript overall is well written and offers new insights in the in vitro antibacterial and antibiofilm activity of Hungarian honeys against respiratory  tract bacteria. 

1) I am puzzled by some figures presented in the manuscript and its originality. For example, in consider Figures 1 and 2. I have never seen figures of these types before. 

2) What is the rationale behind using the Hungarian black locust, linden, and sunflower honey samples? It would be beneficial to have a substantial justification as with regards to its usage or any type of hypothesis that would warrant its effectiveness against respiratory tract bacteria.

3) It is unclear as to how many total honey samples were used

4) I suggest that there is a figure summarizing the procedures/methods because it is a little confusing to go through each step and the particular sequence. 

5) How did the honey producers analyzed and confirmed the botanical origin of honey? How does one ensure the purity of honey and that no adulterants were added?

Reviewer 2 Report

The topic of the article is interesting and the experiments from a microbiological point of view are thorough. What is totally missing is the analysis of honey. The authors are not clear about what honey is. In the discussion they define it as a product of plants (page 11 line 350) collected by bees (page 11 line 382-383). COUNCIL DIRECTIVE 2001/110/EC of 20 December 2001 relating to honey defines honey as follows “Honey is the natural sweet substance produced by Apis mellifera bees from the nectar of plants or from secretions of living parts of plants or excretions of plant-sucking insects on the living parts of plants, which the bees collect, transform by combining with specific substances of their own, deposit, dehydrate, store and leave in honeycombs to ripen and mature.” Numerous researches are going into the investigation of the antibacterial activity of honey, but they are always correlated with physical, chemical and biochemical analyses of the honey in order to discuss the results with its characteristics. Melissopalynological analysis (here called microscopic pollen analysis) is very important but not sufficient.

Minor revision

Title: in vitro must be written in italics

Introduction:

Page 1 and 2 lines 34-35, 36-37, 41-42, 45-47, 47-49, 51-54, 55-57, 58-60, lacking bibliographical references, please add them

Page 2 line 56 please write in full what the abbreviation “hacek” means and put the word “hacek” in brackets

Materials and methods:

why in the different microbiological investigations 3.2.1 (25-50%), 3.3.1 (20-40-60%), 3.3.2 (20-40-60%), were not always the same percentages of honey used? why was 60% honey chosen for the kinetics?

Page 2 line 83 “microscopic pollen analysys” is “Melissopalynological analysis” please give the appropriate name to the analysis.

Page 2 and 3 lines 85-98 please specify the bibliographic reference of the protocol used

Page 2 lines 88 and 99 please change “centrifuged with” with “centrifuged at”.

Page 2 lines 88 and 99 and page 4 line 163 In the paragraph on pollen analysis use rpm to indicate the centrifuge speed and in the paragraph on the study of membrane degradation use g, please standardize throughout the chapter on materials and methods.

Page 3 lines 118-120 I would put this paragraph in the "agar well diffusion assay" section before talking about media and bacteria.

Page 3 line 131 spp. should not be put in italics

pagee 5 figure 1a, are you sure this is a photograph of Robinia pollen grains? It looks more like a photo of a Brassicacea....

page 5 section 3.2.1 please indicate the p values and the test value if the data comparisons are significantly different

In the note to table 2 the authors specify that they interpret the differences in the letters as intra-column significance, but this does not add up. In the column of P. aeruginosa there are two "ac" without a single "c". Furthermore, equal values in the same column have different letters, how is this possible? Please check the statistics again of all the table.

page 6 section 3.2.2 lines 241-254 please indicate the p values and the test value if the data comparisons are significantly different

page 7 fig 3 I suggest putting the letters in descending order of the size of the values and not at random so that you also have an idea of the scale of the values in the graphs. If the statistical analysis was carried out for all 16 theses at the same time, the bonferroni correction is required.

Page 7 and 8 section 3.3.1 please indicate the p values and the test value if the data comparisons are significantly different. In figure 4 there are only results for P. aeruginosa, all other bacteria are missing. Please add a table or figure to fill in the missing data.

page 8 and 9 section 3.3.2 please indicate the p values and the test value if the data comparisons are significantly different

page 9 lines 311-316 this part should be put in the conclusion.

page 9 line 321 table 5 would show 20.3-52.5%, please correct.

Page 10 table 5 are mic value and mic/2 value two replicates? how come they are so different?

page 10 section 3.4.2 please indicate the p values and the test value if the data comparisons are significantly different

page 10 figure 5 I suggest putting the letters in descending order of the size of the values and not at random so that you also have an idea of the scale of the values in the graphs.

Discussion:

Page. 11 line 350 it is essential to change the definition of honey!

Page 11 and 12 lines 350-394, 398-408 this whole part of the discussions would be a part of the introduction please remove or move it

Page 11 line 383 bees do not collect honey but produce it!

Page 11 lines 394-397, page 12 lines 429-432 these values were significantly higher?

Page 12 lines 402-405 please clarify.

Page 12 lines 434-435 bibliographic reference is missing, please add it

Page 12 line 442-445 usually in articles such a statement starts with 'to the best of our knowledge'.

Round 2

Reviewer 1 Report

The authors have addressed the comments. I would suggest changing the colors of the figure as it appears not suited for a publication quality.  

Author Response

Thank you for accepting our responses from the previous round.

Now we have changed the colours and improved the quality of Figure 3.

Reviewer 2 Report

Dear Authors,

I appreciate the great work done by the authors. There are, however, two  minor revisions to be made:
1) in the supplementary material, "vs" should be written in italics because it is a Latin word
2) in table 2 of the manuscript in the columns of P. aeruginosa and S. pneumoniae, the little letters of the statistics still do not fit. How is it possible that there is an "ab" without there being a "b" only? It does not make sense as an indication of statistical difference. It has to be corrected.

Author Response

Thank you for appreciating our efforts to improve the manuscript.

Now we have corrected the two minor points you have indicated:

1) in the supplementary material, "vs" should be written in italics because it is a Latin word

It has been corrected throughout the supplementary tables.

2) in table 2 of the manuscript in the columns of P. aeruginosa and S. pneumoniae, the little letters of the statistics still do not fit. How is it possible that there is an "ab" without there being a "b" only? It does not make sense as an indication of statistical difference. It has to be corrected. 

The superscript letters indicating statistical difference have been revised and corrected (please see Table 2 attached).
